# Recreational screen time and adolescents' school adjustment based on latent profile analysis: The mediating role of perceived physical health

**Jian Yang[1], Yuan Zhang[2], Ming Wu[3], Huiyu Shi[1], Bianjiang Zhang[1], Zhihui Li[1]***

1 College of Physical Education and Health, East China Normal University, Shanghai, China, 2 College of Physical Education and Health, Hunan University of Technology and Business, Changsha, Hunan, China, 3 School of Physical Education (Main Campus), Zhengzhou University, Zhengzhou, Henan, China

* lzh724001437@163.com

## Abstract

This study surveyed 12529 adolescents and employed latent profile analysis to explore the types of adolescents' school adjustment. Multiple linear regression and the Bootstrap method were used to investigate the predictive role of recreational screen time in adolescents' school adjustment and the mediating mechanism of perceived physical health. The results revealed three types of school adjustment in adolescents: "Ideal Type", "Growth Type", and "Ambivalent Type". Recreational screen time was found to significantly and negatively predict both adolescents' school adjustment and their classification into adjustment types. Furthermore, recreational screen time indirectly predicted adolescents' school adjustment through the mediating role of perceived physical health. These findings suggest the importance of appropriately controlling recreational screen time, especially for adolescents in the ideal type category, and further exploring the physical and mental development patterns of adolescents in the ambivalent type, to provide a theoretical basis for improving school adjustment in adolescents.

## Introduction

School adjustment refers to the process through which students actively adjust their physical and mental states in interaction with the school environment to successfully complete the prescribed academic tasks and meet the educational requirements set by the school [1]. As the primary setting for adolescents' daily life, learning, and peer interactions, schools serve as an important platform for socialization and the cultivation of adaptive abilities. The secondary school years are a critical stage for adolescents' physical and mental development, characterized by high levels of academic, social, and emotional pressure. Therefore, the state of school adjustment is crucial for adolescents' overall well-being and academic achievement. Particularly at this stage, students' adaptive capabilities are closely related to their physical and mental

**Data availability statement:** Study data are subject to PHDA access protocols and require registration/approval (Access link: https://www.ncmi.cn/phda/dataDetails.do?id=C-STR:17970.11.A0031.202107.209.V1.0).

**Funding:** This study was supported by funding awarded to Professor Yang Jian from the MOE (Ministry of Education) Foundation on Humanities and Social Sciences [Grant Numbers 22YJA890032]. In his capacity as the funding recipient, Professor Yang Jian contributed to the study design, data collection and analysis, as well as preparation of the manuscript.

**Competing interests:** The authors have declared that no competing interests exist.

health, social skills, and emotional regulation, thus requiring a deeper exploration of the various factors influencing school adjustment and their underlying mechanisms.

Previous studies have indicated that individual factors (such as academic performance [2], emotional experiences [3], and cognitive judgments [4]) and environmental factors (such as family atmosphere [5], teacher-student relationships [6], and peer friendships [7]) are closely related to school adjustment. However, there remain many unknowns regarding the exploration of behavior-related factors. With the increasing frequency of electronic device usage in adolescents' daily lives, screen time has become an important factor influencing their physical and mental health. In particular, recreational screen time, such as watching videos, playing video games, and chatting online—has gradually become a significant part of adolescents' daily routines [8]. In response to growing concerns over the negative effects of screen time, the World Health Organization (WHO) issued guidelines for sedentary behavior among children and adolescents, strongly recommending that children and adolescents limit their recreational screen time to no more than two hours per day [9,10]. Research has shown that longer screen time is closely associated with the physical and mental health of children and adolescents, with excessive recreational screen time significantly linked to negative factors such as poor health, obesity, and mental health problems [11]. Although existing studies have explored the relationship between screen time and school adjustment, most of the research focuses on social media use or general screen time, without specifically addressing recreational screen time as a distinct form of screen behavior, nor providing a detailed analysis of the adolescent population. Based on this, the present study specifically examines the impact of recreational screen time, as a behavioral factor, on adolescents' school adjustment, aiming to provide empirical support for improving adolescents' well-being and physical and mental health development in school settings.

### Research hypothesis

Recreational screen time refers to the total time adolescents spend on screens during leisure activities, and it is an important form of sedentary behavior [8]. In recent years, low physical activity coupled with high screen time has become a common behavioral pattern among adolescents [12]. The increase in screen time is not only associated with adolescents' use of electronic devices to develop academic skills, but also reflects the growing role of online entertainment activities (such as video games and social networking) as an indispensable part of daily life [13]. Specifically, the use of social media has become an integral part of adolescents' daily lives, which may impact their school adjustment. Studies have shown that middle school students who spend longer time on social media tend to have poorer academic performance, worse peer relationships, and a significant negative correlation between social media use and school adjustment [14]. Further longitudinal research has also indicated that the use of social media is significantly negatively correlated with adolescents' behavior, social skills, and emotional functioning [15].

In addition to social media, screen time also negatively impacts adolescents' academic performance and school adjustment. Research on elementary school students

has found that screen time not only directly negatively predicts school adjustment but also indirectly affects school adjustment through academic performance [16]. Moreover, increased screen time has been significantly negatively correlated with children's socio-psychological adjustment and academic performance. One study indicated that Canadian children at 29 months old who spent more time watching television had significantly poorer socio-psychological adjustment and academic performance [17].

From a theoretical perspective, Cognitive Load Theory (CLT) explains the potential impact of prolonged recreational screen time on adolescents' adjustment. This theory posits that all cognitive activities require cognitive resources. Long periods of recreational screen time, especially when engaging in multiple activities simultaneously (such as social networking, gaming, and video browsing), may distract adolescents' attention and create excessive cognitive load. This not only affects their ability to complete academic tasks but also reduces their learning efficiency and mental preparedness, ultimately depleting the resources necessary for school adjustment [18].

In conclusion, screen time negatively impacts adolescents' school adjustment through multiple pathways. Based on the aforementioned theoretical and empirical research, the present study proposes Hypothesis 1: recreational screen time negatively predicts adolescents' school adjustment.

Perceived physical health, also known as self-rated health, refers to an individual's subjective perception of their own health status [19]. As one of the commonly used indicators of physical and mental health [20], it has demonstrated high reliability and validity in predicting a range of health outcomes [21]. Bandura's Triadic Reciprocal Causation Theory posits that there is an interaction between the individual, behavior, and environment [22]. In this context, recreational screen time, as a form of sedentary behavior in adolescents, may influence their perception of their own health. Conversely, changes in perceived physical health can also affect an individual's ability to adjust to the school environment, particularly their social abilities and antisocial behavior.

Research has shown that, compared to children aged 4–12 who watch television less than 3 hours a day, children who watch television for more than 3 hours a day typically report poorer self-rated health and exhibit fewer prosocial behaviors [23]. Further studies have indicated a significant positive correlation between high screen time and adolescents' physical and mental health conditions, such as sleep quality, musculoskeletal pain, and depressive symptoms [24]. Simultaneously, low screen time combined with high physical activity is highly correlated with good self-rated health in adolescents [25]. Thus, excessive screen time may lead to a decline in adolescents' physical health, which in turn affects their health perceptions. When individuals become aware of their health issues, they may experience emotional fluctuations, low self-confidence, and anxiety [26,27], all of which can further impact their adjustment in the school environment [28].

Existing studies have shown a positive correlation between perceived physical health and school adjustment [29], and have suggested that perceived physical health plays a mediating role between screen time and school adjustment [19]. Based on these findings, the present study hypothesizes that recreational screen time may indirectly affect adolescents' school adjustment through its impact on their perception of physical health. Therefore, Hypothesis 2 is proposed: recreational screen time indirectly predicts adolescents' school adjustment through the mediating role of perceived physical health.

Although existing research has revealed the associations between screen time, physical health, and school adjustment, most studies overlook individual differences and tend to analyze the simple relationships between variables. Latent Profile Analysis (LPA), as an individual-centered statistical technique, can identify heterogeneity among individuals and provide an in-depth exploration of the characteristics and behavioral patterns of different groups [30]. Therefore, this study plans to use the LPA method to segment middle school students' school adjustment and, based on this segmentation, explore the predictive role of recreational screen time on school adjustment and its mediating mechanism through perceived physical health. The research model is shown in Fig 1.

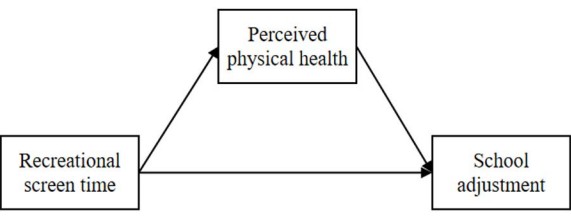

**Fig 1. Research model diagram.**

## Materials and methods

### Data sources

The data for this study is sourced from the National Population Health Data Science Center's Youth Health Topic Database in the Population Health Data Warehouse (PHDA) (website: https://www.ncmi.cn/phda/dataDetails.do?id=C-STR:17970.11.A0031.202107.209.V1.0). The data was collected using a Probability Proportional to Size Sampling (PPS) method and includes multidimensional information on the family situation, physical activity behaviors, school adjustment, quality of life, social interactions, and physical health of middle and high school students. A total of 99,327 students from 186 schools across 17 cities in Shandong Province were surveyed [31]. All participants voluntarily took part in the survey, and both students and their parents signed written informed consent forms. The study was approved by the Shandong University Ethics Committee (20,180,517) [19]. This study uses cross-sectional data from 2020/2017, which merges individual information, family data, school adjustment, and risky behavior datasets. After cleaning the variables and removing missing data, a final sample of 12529 adolescent students was retained. This includes 6388 (50.99%) male students and 6141 (49.01%) female students.

### Variable description

Independent Variable: recreational screen time. This is measured by asking students how much time they spend playing video games and using the computer for non-study-related activities, such as QQ, WeChat, YouTube, etc. The scale ranges from "I do not play video games or use the computer for non-study-related activities" to "5 or more hours per day", with 7 levels in total [32]. A continuous numerical variable, scored from 1 to 7, is generated based on responses to this question.

Dependent Variable: School Adjustment. This is measured using the School Social Behavior Scale (SSBS-2), designed by Merrell and revised by Wang Yan. The scale consists of 65 items, using a 5-point Likert scale, where 1 represents "never occurs" and 5 represents "occurs very frequently". SSBS-2 includes two major aspects: Social Competence and Antisocial Behavior. Social competence comprises three sub-dimensions: peer relation, self-management, and academic behavior. Antisocial behavior includes three sub-dimensions: hostile-irritable, antisocial-aggressive, and defiant-disruptive [33]. The Cronbach's α for the various sub-dimensions are as follows: 0.941, 0.910, 0.889, 0.938, 0.907, and 0.898.

Mediator Variable: Perceived Physical Health. This is assessed by asking students for an overall evaluation of their own health, ranging from "poor" to "excellent", with five levels in total [19,21].

Control Variables: Control variables are divided into individual characteristics and family characteristics. Individual characteristics include: gender, grade level, bullying experience, only child status, and household registration type. Family characteristics include: mother's education level, father's education level, parental relationship, and family economic status. Descriptive statistics for these variables are presented in Table 1.

**Table 1. Descriptive statistics for variables (N = 12529).**

| Variable type | Variable name | Variable Description | Mean value/Percentage |
|---|---|---|---|
| Independent variable | Recreational screen time | 1-7 indicate from no recreational screen time to 5 hours and above, respectively | 2.06 |
| Dependent variable | School adjustment | 65 entries worth 1–5 points each | 270.85 |
| Mediator variable | Perceived physical health | 1-5 indicate the level of health from poor to good, respectively | 3.72 |
| Individual characteristic | Gender | 0 = female; 1 = male | 49.01%/50.99% |
| | School year | 0 = middle school; 1 = high school | 61.66%/38.34% |
| | Bullying | 0 = have not experienced bullying; 1 = have experienced bullying | 87.20%/12.80% |
| | Only child | 0 = No; 1 = Yes | 70.62%/29.38% |
| | Household type | 1 = rural; 2 = urban; 3 = residential | 58.46%/23.27%/18.28% |
| Family characteristics | Mother's education level | 1-9 from lowest to highest representing no education to master's degree and above | 4.00 |
| | Father's education level | 1-9 from lowest to highest representing no education to master's degree and above | 4.30 |
| | Parental relationship | 0 = bad; 1 = good | 15.04%/84.96% |
| | Family economic level | 1-5 indicate low to high economic level, respectively. | 2.90 |

## Analytical methods

This study uses STATA 17.0 for data merging, variable processing, sample exclusion, and descriptive statistics. Subsequently, Mplus 8.3 is employed for Latent Profile Analysis (LPA) of school adjustment to identify different distributions and types of school adjustment within the adolescent population, and to explore the relationship between recreational screen time and school adjustment profiles, along with the mediating mechanisms involved. Specifically, the data processing and analysis are divided into two phases.

In the first stage, LPA analysis was conducted to determine the optimal number of categories of school adjustment for the adolescent group based on the AIC, BIC, aBIC, Entropy, LMR, BLRT indicators and the practical implications of the profiles.

In the second phase, the obtained school adjustment profile results are treated as the dependent variable, and multiple linear regression is performed to examine the impact of recreational screen time on school adjustment across different profiles.

Finally, to investigate the mediating mechanism of the influence of recreational screen time on school adjustment, recreational screen time is treated as the independent variable, perceived physical health as the mediator, and school adjustment as the dependent variable. A Bootstrap mediation analysis is conducted using Model 4 in the Process Plugin of SPSS 26.0. The significance level is set at $\alpha = 0.05$.

## Results

### Latent profile analysis of adolescents' school adjustment

Mplus 8.3 software was used to perform Latent Profile Analysis (LPA) on adolescents' school adjustment. The school adjustment categories were sequentially set as Category I, Category II, Category III, Category IV, Category V, and Category VI. Model comparison was conducted based on fit indices. Generally speaking, when the LMR and BLRT are significant, the smaller the values of AIC, BIC, and aBIC, and the closer the Entropy value is to 1, the better the model fit. An Entropy value greater than 0.80 indicates a classification accuracy higher than 90%. If the LMR and BLRT values are

smaller than 0.05, it suggests that the model with k categories fits significantly better than the model with k-1 categories [34]. The fit indices for each category are presented in Table 2.

As shown in Table 2, all latent profile models for adolescents' school adjustment exhibit good model fit. Therefore, following the approach of VAZIRI [35], the BIC inflection point chart was plotted to further determine the optimal number of profiles (see Fig 2). From the chart, it is evident that the BIC curve begins to flatten from Category III, and the slope of the curve does not increase with the addition of more categories. Moreover, Category III also satisfies the recommendation from previous studies that each profile should constitute no less than 5% of the total sample [36,37]. Based on this comprehensive assessment, the study concludes that Category III is the optimal latent profile model.

The proportions of the three latent profiles of school adjustment and their mean scores on six dimensions are shown in Fig 3. Category 1 accounts for 25.6% of the sample. Adolescents in this category have mean scores on the three social abilities (peer relation, self-management, and academic behavior) ranging from 23.5 to 37.7, and their mean scores on the three antisocial behaviors (hostile-irritable, antisocial-aggressive, and defiant-disruptive) range from 12.8 to 18.8. The mean scores for social abilities in this category are higher than for antisocial behaviors, but there is still considerable room

**Table 2. Fit indices for latent profile analysis of adolescent school adjustment.**

|  | AIC | BIC | aBIC | Entropy | LMR(P) | BLRT(P) | Group size |
|---|---|---|---|---|---|---|---|
| I | 523907.313 | 523996.543 | 523958.408 | — | — | — | — |
| II | 479940.145 | 480081.425 | 480021.045 | 0.990 | <0.001 | <0.001 | 0.869/0.131 |
| III | **462251.616** | **462444.947** | **462362.322** | **0.921** | **<0.001** | **<0.001** | **0.256/0.114/0.630** |
| IV | 453293.449 | 453538.830 | 453433.960 | 0.937 | <0.001 | <0.001 | 0.254/0.112/0.626/0.008 |
| V | 443249.070 | 443546.502 | 443419.387 | 0.936 | <0.001 | <0.001 | 0.600/0.195/0.107/0.091/0.008 |
| VI | 435338.483 | 435687.966 | 435538.605 | 0.917 | <0.001 | <0.001 | 0.050/0.096/0.358/0.399/0.090/0.008 |

Note: Bolded categories are optimal models.

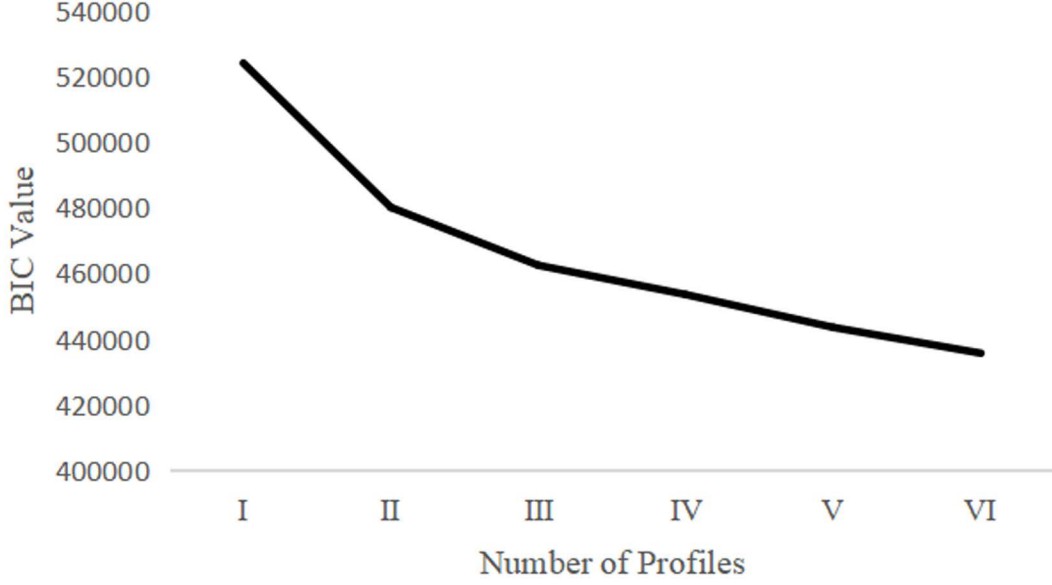

**Fig 2. Inflection point diagram of BIC values.**

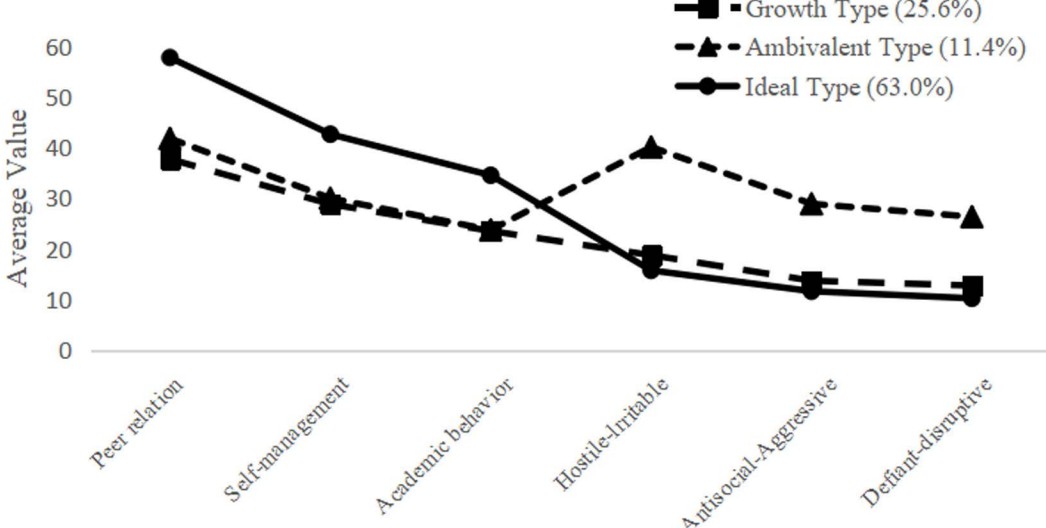

**Fig 3. Profiles of adolescent school adjustment III categories.**

for improvement in social abilities and a potential decrease in antisocial behaviors. This indicates developmental potential, so the school adjustment profile of this category is named "Growth Type".

Category 2 accounts for 11.4%. Adolescents in this category have higher mean scores on the three social abilities (23.7 to 41.8) than those in Category 1, yet their mean scores on the three antisocial behaviors are significantly higher than in the other two categories (26.3 to 40.1). This indicates a higher level of negative traits. Given the similar mean scores for both social abilities and antisocial behaviors, the school adjustment profile of this category is named "Ambivalent Type".

Category 3 accounts for 63.0%. Adolescents in this category have the highest mean scores on the three social abilities (34.5 to 57.8) across the three categories, and their mean scores on the three antisocial behaviors are the lowest (10.2 to 15.8). Overall, this group demonstrates high social abilities and low antisocial behaviors, aligning with expectations for adolescents' school adjustment. Therefore, this school adjustment type is named "Ideal Type".

To further examine the accuracy of the latent profile classification results, this study conducted a difference test on the six sub-dimensions of school adjustment. The results are shown in Table 3. As can be seen from the table, the three latent profiles "Growth Type", "Ambivalent Type", and "Ideal Type" show significant differences across the six sub-dimensions, indicating the distinctiveness of each category. The results of the analysis of variance (ANOVA) show that the "Growth Type" school adjustment profile has significantly lower scores on the social ability dimension compared to the other two profiles. The "Ambivalent Type" school adjustment profile has significantly higher scores on the antisocial behavior dimension compared to the other two profiles. The "Ideal Type" school adjustment profile, in contrast, has significantly higher scores on the social ability dimension compared to both the "Growth Type" and "Ambivalent Type" profiles, and significantly lower scores on the antisocial behavior dimension compared to both the "Growth Type" and "Ambivalent Type" profiles. These findings confirm the reliability of the naming results based on the mean scores of the different latent profile categories, and also highlight the heterogeneity in adolescents' school adjustment status.

### Regression analysis

Multivariate logistic regression was conducted using the three latent profile categories as the dependent variable and recreational screen time as the independent variable. The "Growth Type" profile was set as the reference group to examine the impact of recreational screen time on school adjustment types. The regression results are shown in Table 4. The study found

**Table 3. Comparison of mean differences between latent profile of school adjustment across six dimensions.**

| Variable | Growth Type | Ambivalent Type | Ideal Type | F | Multiple comparisons |
|---|---|---|---|---|---|
| Peer relation | 37.47±8.99 | 41.78±9.64 | 57.80±7.87 | 7611.27*** | 1<2<3 |
| Self-management | 28.65±6.66 | 30.00±7.00 | 42.66±4.73 | 8710.09*** | 1<2<3 |
| Academic behavior | 23.38±5.83 | 23.72±6.10 | 34.54±3.85 | 8005.84* | 1<2<3 |
| Hostile-irritable | 18.79±4.56 | 40.10±8.73 | 15.75±2.56 | 19882.13*** | 3<1<2 |
| Antisocial-aggressive | 13.74±3.38 | 28.90±6.16 | 11.62±2.04 | 18355.85*** | 3<1<2 |
| Defiant-disruptive | 12.78±3.48 | 26.33±5.57 | 10.23±1.99 | 17209.44*** | 3<1<2 |

Note: P<0.05*,P<0.01**,P<0.001***.

**Table 4. Multivariate logistic regression of the effect of screen time for recreation on type of school adjustment.**

| Independent variable | Class | B | OR | 95%CI |
|---|---|---|---|---|
| Recreational screen time | Ambivalent Type | 0.075*** | 1.077 | [1.041, 1.115] |
| | Ideal Type | −0.248*** | 0.780 | [0.760, 0.802] |

Note: Growth Type as the reference group; P<0.05*,P<0.01**,P<0.001***.

that adolescents with higher recreational screen time were more likely to be categorized into the "Ambivalent Type" profile, whereas adolescents with lower recreational screen time were more likely to be categorized into the "Ideal Type" profile.

Using recreational screen time as the independent variable and the different latent profile categories of school adjustment as the dependent variables, along with control variables for individual and family characteristics, four multivariate linear regression models were constructed to explore the specific predictive effects of recreational screen time on school adjustment in different adolescent profiles. The regression results are shown in Table 5.

**Table 5. Multiple linear regression results for recreational screen time and adolescent school adjustment.**

| Variable | Model 1 | Model 2 | Model 3 | Model 4 |
|---|---|---|---|---|
| Recreational screen Time | −0.708** (0.207) | −0.138 (0.305) | −1.279*** (0.149) | −4.200*** (0.205) |
| Gender (0=female) | −4.991*** (0.710) | 1.032 (1.289) | 0.880* (0.381) | −5.773*** (0.625) |
| School year (0=middle school) | 2.441** (0.739) | 0.764 (1.447) | −0.454 (0.382) | 6.216*** (0.638) |
| Bullying(0=have not experienced bullying) | −11.198*** (0.981) | −6.126*** (1.225) | −2.065** (0.770) | −30.776*** (0.943) |
| Only child (0=No) | −4.151*** (0.835) | −2.655* (1.248) | 0.805 (0.456) | −7.350*** (0.721) |
| Household type (rural) | | | | |
| Urban | −0.774 (1.020) | 1.619 (1.587) | −0.198 (0.513) | 1.984* (0.843) |
| Residential | −7.740*** (0.897) | −0.045 (1.548) | 0.652 (0.532) | −4.397*** (0.837) |
| Mother's education level | 0.424 (0.253) | 0.174 (0.315) | 0.343* (0.135) | 0.821*** (0.211) |
| Father's education level | 0.593* (0.247) | 0.521 (0.327) | 0.334* (0.133) | 1.565*** (0.209) |
| Parental relationship | 6.927*** (0.886) | 1.666 (1.289) | 3.280*** (0.671) | 21.824*** (0.889) |
| Family economic level | 0.204 (0.561) | 1.534*** (0.556) | 2.525*** (0.382) | 3.348*** (0.495) |
| Constant term | 240.289*** (1.907) | 193 298*** (2.394) | 283.982*** (1.308) | 248.217*** (1.721) |
| N | 3170 | 1429 | 7930 | 12529 |

Note: P<0.05*,P<0.01**,P<0.001***; coefficients followed by standard error.

Model 1 used the "Growth Type" school adjustment profile as the dependent variable. The results indicated that recreational screen time significantly negatively predicted school adjustment in this profile ($\beta = -0.708$, $P < 0.05$).

Model 2 used the "Ambivalent Type" school adjustment profile as the dependent variable. The results showed that recreational screen time still negatively predicted school adjustment, but this effect was not statistically significant ($\beta = -0.138$, $P > 0.05$).

Model 3 used the "Ideal Type" school adjustment profile as the dependent variable. The results revealed that recreational screen time significantly negatively predicted school adjustment in this profile ($\beta = -1.279$, $P < 0.05$), and the predictive effect was stronger than in the "Growth Type" and "Ambivalent Type" profiles.

Model 4 conducted regression analysis with school adjustment for the entire sample as the dependent variable. The results showed that recreational screen time significantly negatively predicted adolescents' school adjustment ($\beta = -4.200$, $P < 0.05$), with the predictive effect being significantly higher than in the first three models.

## Mediation analysis

To systematically test the model fit between the research model and the observed data, Structural Equation Modeling (SEM) was conducted using AMOS. After incorporating the independent variable, mediating variable, dependent variable, and control variables, it was found that Model 1, Model 3, and Model 4 had good model fit indices (although CMIN/DF was slightly large, the results were still acceptable considering the large sample size). However, since the TLI value of Model 2 was much lower than 0.9, its fit indices were not ideal. Detailed results are shown in Table 6.

Subsequently, the Bootstrap extraction was set to 5000 times in SPSS 26.0 using the Process plugin for mediation effect testing. The results are shown in Table 7. The study found that recreational screen time can predict school adjustment in Models 1, 3, and 4 through the mediation effect of perceived physical health (95% CI: −0.044, −0.012; 95% CI: −0.060, −0.038; 95% CI: −0.179, −0.148), but it cannot predict school adjustment in Model 2 through perceived physical health (95% CI: −0.028, 0.018).

## Discussion

### Different profiles of adolescents' school adjustment

Through profile analysis and subsequent difference testing of school adjustment among middle school students, it was found that there is heterogeneity in the school adjustment of this group. Based on the identified profile types, they can be named as "Growth Type", "Ambivalent Type", and "Ideal Type", which is consistent with previous research findings [38,39]. According to Erikson's theory of psychosocial development, whether adolescents can balance their sense of identity and role confusion during middle school is crucial to their mental health and social adaptability [40].

In this study, 63% of middle school students were classified as the "Ideal Type". This group had the highest scores in the social ability dimension and the lowest scores in the antisocial behavior dimension, indicating that the majority of middle school students have good school adjustment and are able to reasonably manage developmental contradictions. This

**Table 6. Model fitting indicators.**

| Model Type | CMIN/DF | CFI | TLI | GFI | RMSEA |
|---|---|---|---|---|---|
| Model 1 | 4.976 | 0.979 | 0.918 | 0.996 | 0.035 |
| Model 2 | 4.719 | 0.944 | 0.780 | 0.992 | 0.051 |
| Model 3 | 6.942 | 0.990 | 0.959 | 0.998 | 0.027 |
| Model 4 | 11.798 | 0.990 | 0.959 | 0.998 | 0.029 |

**Note**: Model 1 represents the "Growth Type", Model 2 represents the "Ambivalent Type", Model 3 represents the "Ideal Type", and Model 4 represents the overall type.

**Table 7. Test results of the mediating effect of perceived physical health.**

| Model type | | Standardized effect value | S.E | 95%CI |
|---|---|---|---|---|
| Model 1 | Total effect | −0.028 | 0.008 | [-0.044, -0.012] |
| | Direct effect | −0.031 | 0.008 | [-0.047, -0.016] |
| | Indirect effect | 0.003 | 0.001 | [0.001, 0.006] |
| Model 2 | Total effect | −0.005 | 0.012 | [-0.028, 0.018] |
| | Direct effect | −0.005 | 0.012 | [-0.028, 0.018] |
| | Indirect effect | −0.000 | 0.001 | [-0.002, 0.002] |
| Model 3 | Total effect | −0.049 | 0.006 | [-0.060, -0.038] |
| | Direct effect | −0.040 | 0.006 | [-0.051, -0.030] |
| | Indirect effect | −0.009 | 0.002 | [-0.012, -0.005] |
| Model 4 | Total effect | −0.164 | 0.008 | [-0.179, -0.148] |
| | Direct effect | −0.151 | 0.008 | [-0.167, -0.136] |
| | Indirect effect | −0.012 | 0.002 | [-0.016, -0.010] |

**Note**: Model 1 represents the "Growth Type", Model 2 represents the "Ambivalent Type", Model 3 represents the "Ideal Type", and Model 4 represents the overall type.

also aligns with the standards of school education, which aim to cultivate students with good interpersonal relationships, certain self-management skills, academic proficiency, and a gentle character, without significant antisocial tendencies [41].

Secondly, 25.6% of students were classified as the "Growth Type". These students had the lowest scores in the social ability dimension across all profiles, but their antisocial behavior was close to that of the "Ideal Type". This group exhibits good behavioral norms but lacks social skills to some extent, more closely resembling the traditional "obedient students". Therefore, as the "reserve force" for the "Ideal Type" of school adjustment, it is necessary to strengthen the development of social skills for the "Growth Type" students [42].

Finally, 11.4% of students were classified as the "Ambivalent Type". These students had slightly higher scores in the social ability dimension compared to the "Growth Type" but exhibited significantly higher antisocial behavior than the other two types, showing clear "rebellious" characteristics. This group is often referred to as "problem students" or "students in need of improvement" in school education. They face significant phase-related psychological crises, and addressing the challenges of this group is an important task throughout their school education [43].

In summary, the school adjustment of adolescents largely meets expectations, with a small portion requiring further development of social skills, and a very small number of students exhibiting concerning school adjustment. Therefore, further exploration of the influencing factors of different school adjustment profiles is an important measure to comprehensively improve the well-being of middle school students in school life.

### The predictive role of recreational screen time on adolescents' school adjustment

Taking school adjustment type as the dependent variable, the study found that recreational screen time significantly negatively predicts the school adjustment types of middle school students. From the internal changes of different school adjustment types, recreational screen time significantly increases the probability of adolescents with "Growth Type" school adjustment being classified into "Ambivalent Type" by 7.7%, while the probability of being classified into "Ideal Type" decreases by 22%. This suggests that recreational screen time not only negatively predicts school adjustment directly but also differentiates the expression of individuals in different school adjustment types.

Further analysis of the predictive role of recreational screen time on specific types of school adjustment revealed that, except for the "Ambivalent Type" students, where the predictive effect was not significant, recreational screen time significantly predicts "Growth Type", "Ideal Type", and overall school adjustment. Among these, the impact of recreational

screen time on overall school adjustment is the strongest, followed by the "Ideal Type", and lastly the "Growth Type". This indicates that the effect of recreational screen time on individuals with good school adjustment is much greater than that on individuals with moderate school adjustment. In short, compared to other levels of school adjustment, middle school students with better school adjustment are more adversely affected by the same amount of recreational screen time.

Previous studies have found that excessive screen time among adolescents may lead to various adverse effects, such as impaired cognitive and socio-psychological development, increased risk-taking behaviors (such as substance abuse, alcohol consumption, and suicidal tendencies), as well as health problems like obesity, depression, and sleep disturbances [44,45]. These factors can interfere with adolescents' school adjustment in multiple ways. Specifically, screen time used for entertainment purposes (e.g., online socializing, gaming) is closely associated with internet addiction [46]. Social media has become the primary source of entertainment for middle school students, potentially leading them to neglect responsibilities, spend excessive time online, and experience anxiety and compulsive symptoms when they are unable to access these platforms [47].

Media Dependency Theory suggests that the more services a medium provides, the greater the dependency of its audience and society on it. When individuals excessively rely on a medium to meet their needs, the use of that medium may have negative effects on them [48,49]. Recreational screen time, which relies on media such as computers or mobile phones, can also lead to adverse outcomes when middle school students develop excessive dependence. Studies have shown a significant correlation between mobile phone dependency, mental health, and school adjustment [50], and mobile phone dependency is a potential influencing factor for poor school adjustment [51]. Therefore, mobile phone dependence and internet addiction may lead adolescents to experience academic procrastination, social anxiety, and other academic and interpersonal relationship issues, which, in turn, affect their school adjustment [52,53]. The duration of recreational screen time serves as an external manifestation of mobile phone dependence and internet addiction.

Furthermore, from a practical perspective, daily time resources are limited. Improving school adjustment means that more time can be dedicated to learning various skills (e.g., academic subjects, interpersonal communication), while minimizing factors that may lead to antisocial behaviors. Recreational screen time, however, erodes learning time resources and increases the risk of exposure to harmful online content. Based on these factors, it is reasonable to conclude that recreational screen time negatively predicts adolescents' school adjustment.

## Mediating role of perceived physical health

This study found that recreational screen time negatively predicts school adjustment through the mediating role of perceived physical health. Specifically, recreational screen time not only predicts adolescents' school adjustment type (e.g., "Ideal Type", "Growth Type") through perceived physical health but also further predicts individual school adjustment types ("Ideal Type", "Growth Type") through this mediating variable. However, it is noteworthy that recreational screen time does not predict the school adjustment of adolescents with a "Ambivalent Type" school adjustment profile through the mediating role of perceived physical health. Combined with the results of the regression analysis, this suggests internal heterogeneity in adolescents' school adjustment.

Overall, recreational screen time can predict the school adjustment of nearly 90% of middle school students through perceived physical health. However, the influencing factors of school adjustment for adolescents with an "Ambivalent Type" school adjustment profile still need further exploration.

The Health Belief Model suggests that individuals' perception of health risks influences their behavior [54]. The increase in recreational screen time significantly impacts both physiological and psychological health, such as obesity, hypertension, poor stress regulation (e.g., sympathetic nervous system activation and cortisol imbalance), internalizing and externalizing behaviors, depressive symptoms, and suicidal tendencies [55]. Perceived physical health, as a key predictive indicator of health, serves as the "canary in the coal mine" for an individual's physical well-being. Therefore, the deterioration in both physical and mental health caused by excessive recreational screen time will be among the first to manifest.

Other studies have demonstrated that poorer physical and mental health is closely associated with lower social support, worse academic adaptability, and higher levels of antisocial behaviors [56–58], which may contribute to the worsening of adolescents' school adjustment from multiple dimensions.

It is worth mentioning that in this study, recreational screen time cannot predict the school adjustment of adolescents in the "Ambivalent Type" profile through perceived physical health. This is directly related to the fact that recreational screen time does not predict the school adjustment of adolescents in the "Ambivalent Type" profile. Specifically, the impact of excessive recreational screen time on the school adjustment of "Ambivalent Type" adolescents may not be solely reflected through changes in perceived physical health. The increase in recreational screen time may not indirectly promote school adjustment by improving their physical and mental health, and may even exacerbate negative behaviors and emotional issues. Conversely, the adaptation difficulties of this group may stem more from individual factors such as emotional regulation, social support systems, and behavioral control, which may not have as direct a relationship with recreational screen time as seen in other groups. Therefore, future research should further explore the unique mechanisms of school adjustment in "Ambivalent Type" adolescents. In addition to recreational screen time and perceived physical health, factors such as emotional regulation, behavioral control, and social support may have a more direct impact on their school adjustment. Interventions for this group may need to focus more on their emotional management and behavioral adjustments to help them better regulate their emotions, reduce antisocial behaviors, and improve school adjustment. Furthermore, future research should incorporate more variables to explore how these factors interact, in order to provide more precise intervention strategies for the "Ambivalent Type" adolescent group. This will help in understanding the school adjustment mechanisms of this group more comprehensively and provide theoretical foundations for improving their physical and mental development.

### Research limitations

This study, taking into account individual differences between variables, uses an individual-centered research approach to categorize school adjustment types and more thoroughly examines the impact of recreational screen time on adolescent school adjustment and its mechanisms. However, several issues remain:

First, despite having a large sample size, this study relies on cross-sectional data, which limits the ability to explore the changes in adolescent school adjustment profiles over time and the causal effects of recreational screen time in predicting school adjustment. Future studies could use longitudinal data to further enhance the validity of the research.

Second, the research data is based on self-reports from adolescents, which may increase the risk of result bias. Future studies could include reports from multiple groups, such as teachers and parents, for comparison, further strengthening the credibility of the findings.

### Conclusion

Adolescent school adjustment exhibits heterogeneity and can be categorized into three types: "Ideal Type", "Growth Type", and "Ambivalent Type". Recreational screen time can significantly and negatively predict the level of school adjustment and the probability of belonging to different categories. Specifically, the longer the duration of entertainment screen use, the more likely individuals are to be classified as "Growth Type" or "Ambivalent Type", moving further away from the "Ideal Type". Furthermore, recreational screen time indirectly influences school adjustment performance through the mediating mechanism of perceived physical health. Specifically, excessive entertainment screen use weakens adolescents' subjective evaluations of their health, reducing their psychological vitality and physical confidence, which negatively affects their self-adjustment and social interaction abilities in the school environment. This study not only expands the theoretical perspective of adolescent school adjustment classification but also reveals the impact pathways of digital lifestyles on school adjustment. Based on these findings, the following recommendations are proposed:

Family Level: Parents are encouraged to strengthen the daily management of adolescents' entertainment screen use, especially paying attention to the duration, usage context, and content type. They should help adolescents develop good media usage habits and health awareness. Additionally, parents should create more opportunities for parent-child interaction, outdoor exercise, and structured learning, in order to increase chances for positive feedback in real life and promote the development of social skills.

School Level: Schools can use psychological assessments and behavioral observations to identify "Growth Type" and "Ambivalent Type" students. Tailored intervention programs, such as social skills training, conflict management courses, and emotional regulation training, should be implemented to enhance students' adaptability. Schools should also emphasize health literacy and media literacy education, helping students establish an integrated cognitive system encompassing physical, psychological, and social health, to resist the adverse effects of excessive screen use on health and interpersonal functioning.

Policy and Societal Level: Educational management institutions should implement more specific guidelines for adolescents' entertainment screen use, strengthening the awareness of "digital citizenship". They should also encourage universities and research institutions to explore broader factors that contribute to improving school adjustment, especially for "Ambivalent Type" adolescents. Additionally, efforts should be made to promote collaboration among families, schools, and communities to build a support network for school adjustment in the context of digital media, enhancing adolescents' self-management and social adaptation abilities in a complex media ecology.

## Acknowledgments

Thanks to the Population Health Data Archive(PHAD) for the Database of Youth Health and all respondents to this project.

## Author contributions

**Conceptualization:** Jian Yang, Yuan Zhang, Zhihui Li.

**Data curation:** Yuan Zhang.

**Formal analysis:** Zhihui Li.

**Methodology:** Ming Wu, Zhihui Li.

**Project administration:** Jian Yang.

**Software:** Huiyu Shi.

**Validation:** Huiyu Shi, Bianjiang Zhang.

**Visualization:** Ming Wu, Bianjiang Zhang.

**Writing – original draft:** Jian Yang, Yuan Zhang, Huiyu Shi, Zhihui Li.

**Writing – review & editing:** Jian Yang, Ming Wu, Bianjiang Zhang, Zhihui Li.

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
