## [Decision Letter · Decision Letter 0]

8 Jun 2025

Dear Dr. Li,

Thank you for submitting your manuscript to PLOS ONE. After careful consideration, we feel that it has merit but does not fully meet PLOS ONE’s publication criteria as it currently stands. Therefore, we invite you to submit a revised version of the manuscript that addresses the points raised during the review process.

**ACADEMIC EDITOR: **

The different pattern observed for the "ambivalent" type regarding the mediating role of perceived physical health suggests an area for deeper investigation. The manuscript points out that recreational screen time did not predict school adjustment for this particular subgroup through the mediating effect of perceived physical health, and notes that the "physical and psychological developmental patterns and influencing factors of ambivalent school-adjusted adolescents need to be further explored". This highlights that while the study successfully identified this distinct subgroup, understanding the specific mechanisms affecting their school adjustment remains less clear, pointing towards a necessary direction for more nuanced future research specifically focused on this particular profile to uncover the unique factors at play. This isn't a flaw in the current study's execution but an acknowledged gap in understanding revealed by its findings, suggesting valuable avenues for future work that would improve upon the current study's scope in this specific area. Please, make a discussion by expressing these facts and ideas for future research.

We look forward to receiving your revised manuscript.

Kind regards,

Javier Fagundo-Rivera, PhD

Academic Editor

PLOS ONE

Journal Requirements:

MOE (Ministry of Education) Foundation on Humanities and Social Sciences [grant numbers 22YJA890032].

5. Your abstract cannot contain citations. Please only include citations in the body text of the manuscript, and ensure that they remain in ascending numerical order on first mention.

6. Please update your submission to use the PLOS LaTeX template. The template and more information on our requirements for LaTeX submissions can be found at http://journals.plos.org/plosone/s/latex.

Additional Editor Comments:

Dear Authors

My comments and those of 2 reviewers are being sent.

I recommend Major Revisions.

Reviewers' comments:

Reviewer's Responses to Questions

**Comments to the Author**

1. Is the manuscript technically sound, and do the data support the conclusions?

Reviewer #1: Yes

Reviewer #2: Yes

2. Has the statistical analysis been performed appropriately and rigorously?

Reviewer #1: Yes

Reviewer #2: Yes

3. Have the authors made all data underlying the findings in their manuscript fully available?

Reviewer #1: Yes

Reviewer #2: Yes

4. Is the manuscript presented in an intelligible fashion and written in standard English?

Reviewer #1: No

Reviewer #2: Yes

**Reviewer #1:**

1.The Introduction should primarily provide background information, clearly identify the problem being addressed, and then outline the research objectives, questions, and structure of the paper. Research hypotheses should not be included in this section. It is recommended that the authors create a separate section for the research hypotheses and supplement this part with additional relevant literature.

2.The authors are also advised to include a model diagram of the research hypotheses. This will help present the study’s hypotheses and theoretical framework in a clearer and more intuitive way.

3.By dividing the adolescent sample into three groups and using multiple regression to test both the direct effect of leisure screen time on school adjustment and the mediating role of perceived physical health, the authors offer initial insights into group differences. However, this approach does not account for self-report measurement error and treats latent profile classification and path analysis as separate, overlooking classification uncertainty and the hierarchical structure of the data. It is recommended that the authors use structural equation modeling (SEM) to systematically examine the measurement models and path relationships among the key latent variables—leisure screen time, school adjustment, and perceived physical health. Afterwards, multiple regression analysis can be used to reveal differences in performance among the latent groups.

4.In the references section, some journal names are abbreviated while others are written in full. The authors are advised to standardize the journal name formatting according to the journal’s citation guidelines to enhance the consistency and professionalism of the references.

5.There are also some minor errors in the manuscript. The authors are encouraged to carefully review the entire paper and make necessary corrections to further improve its clarity and overall quality.

**Reviewer #2: **

Many thanks for your invitation to review the manuscript titled "Recreational screen time and adolescent school adjustment based on latent profile analysis: the mediating role of perceived physical health." Overall, the manuscript topic is interesting and presents valuable findings for the adolescent group. However, the manuscript requires minor revisions before its acceptance.

In the result section, it is stated that "overall sample category should not be less than 5% [32]", the authors may also add a primary source reference with the existing one.

Overall, minor grammatical mistakes need to be addressed.

The conclusion is way too short. The authors could provide additional recommendations to support the target group who might benefit from the study's findings, such as the study's potential for home environment approaches or school administration actions.

**Do you want your identity to be public for this peer review?** For information about this choice, including consent withdrawal, please see our Privacy Policy

Reviewer #1: No

Reviewer #2: No

---

## [Author Response · Author response to Decision Letter 1]

13 Jul 2025

Dear Editor/Reviewer:

Thank you very much for your interest and recognition of this study and for giving us the opportunity to revise and improve "Recreational screen time and adolescent school adjustment based on latent profile analysis: the mediating role of perceived physical health" (ID:PONE-D-25-21675). Your valuable suggestions have made an important contribution to the quality improvement of the manuscript, which we have taken seriously and have been substantially revised and improved. We sincerely hope that the manuscript will be published in Plos One.

This letter is followed by the editor/reviewers' comments and our point-by-point response to the reviewers' comments. Changes in the manuscript are marked using highlight mode. The revision was developed in consultation with all co-authors and each author approved the revision in its final form.

Thank you again for your time and advice.We look forward to hearing from you at your earliest convenience

Sincerely yours,

Jian Yang, Ph.D

ACADEMIC EDITOR: 

The different pattern observed for the "ambivalent" type regarding the mediating role of perceived physical health suggests an area for deeper investigation. The manuscript points out that recreational screen time did not predict school adjustment for this particular subgroup through the mediating effect of perceived physical health, and notes that the "physical and psychological developmental patterns and influencing factors of ambivalent school-adjusted adolescents need to be further explored". This highlights that while the study successfully identified this distinct subgroup, understanding the specific mechanisms affecting their school adjustment remains less clear, pointing towards a necessary direction for more nuanced future research specifically focused on this particular profile to uncover the unique factors at play. This isn't a flaw in the current study's execution but an acknowledged gap in understanding revealed by its findings, suggesting valuable avenues for future work that would improve upon the current study's scope in this specific area. Please, make a discussion by expressing these facts and ideas for future research.

Response:Thank you for your thorough review of our paper and your valuable feedback. Your comments have been extremely helpful in further refining our research and improving the quality of the paper. In response to your suggestion regarding the study of the “Ambivalent Type” adolescent group, we have had in-depth discussions and made revisions. We hope the following explanations and additions can address your concerns.

In our study, we indeed found that recreational screen time significantly predicts the school adjustment of adolescents in the “Ideal Type” and “Growth Type” categories, with perceived physical health playing an intermediary role in this process. However, for the “Ambivalent Type” group, recreational screen time not only fails to directly predict their school adjustment but also cannot indirectly predict their school adjustment through perceived physical health as a mediating factor. This result highlights the particularities of the “Ambivalent Type” adolescent group in terms of school adjustment. Despite performing better in social capabilities such as interpersonal relationships, self-management, and academic skills compared to the “Growth Type”, the difficulties related to antisocial behavior may lead to different impacts on their school adjustment.

We speculate that for “Ambivalent Type” adolescents, the relationship between recreational screen time and school adjustment may not be as direct as in other groups. Due to their higher tendency towards antisocial behavior, the increase in recreational screen time may not promote school adjustment through improved physical health or emotional status. On the contrary, it may exacerbate negative emotions and behaviors. Therefore, future research should further explore the specific mechanisms of school adjustment in “Ambivalent Type” adolescents, particularly focusing on the roles of emotional regulation, behavioral control, and other factors.

We agree with your observation that our current understanding of the school adjustment mechanisms for the “Ambivalent Type” group remains limited. Further exploration in this area is crucial for refining the theoretical framework of adolescent school adjustment. Future studies should consider more psychosocial variables and integrate factors such as emotional regulation, behavioral control, and social support to gain a deeper understanding of the adaptation process for this group. As a result, we have expanded the discussion section of our paper based on your valuable suggestions. The additional content is as follows:

It is worth mentioning that in this study, recreational screen time cannot predict the school adjustment of adolescents in the “Ambivalent Type” profile through perceived physical health. This is directly related to the fact that recreational screen time does not predict the school adjustment of adolescents in the “Ambivalent Type” profile. Specifically, the impact of excessive recreational screen time on the school adjustment of “Ambivalent Type” adolescents may not be solely reflected through changes in perceived physical health. The increase in recreational screen time may not indirectly promote school adjustment by improving their physical and mental health, and may even exacerbate negative behaviors and emotional issues. Conversely, the adaptation difficulties of this group may stem more from individual factors such as emotional regulation, social support systems, and behavioral control, which may not have as direct a relationship with recreational screen time as seen in other groups. Therefore, future research should further explore the unique mechanisms of school adjustment in “Ambivalent Type” adolescents. In addition to recreational screen time and perceived physical health, factors such as emotional regulation, behavioral control, and social support may have a more direct impact on their school adjustment. Interventions for this group may need to focus more on their emotional management and behavioral adjustments to help them better regulate their emotions, reduce antisocial behaviors, and improve school adjustment. Furthermore, future research should incorporate more variables to explore how these factors interact, in order to provide more precise intervention strategies for the “Ambivalent Type” adolescent group. This will help in understanding the school adjustment mechanisms of this group more comprehensively and provide theoretical foundations for improving their physical and mental development. (See lines 444-467)

In the third suggestion, “Policy and Societal Level: Educational management institutions should implement more specific guidelines for adolescents' entertainment screen use, strengthening the awareness of “digital citizenship”. They should also encourage universities and research institutions to explore broader factors that contribute to improving school adjustment, especially for “Ambivalent Type” adolescents. Additionally, efforts should be made to promote collaboration among families, schools, and communities to build a support network for school adjustment in the context of digital media, enhancing adolescents' self-management and social adaptation abilities in a complex media ecology”. (See lines 513-521)

Thank you again for your valuable suggestions. We hope to make up for the deficiencies in the current research through more detailed discussions. We also look forward to your further opinions and guidance.

Journal Requirements:

1.Please ensure that your manuscript meets PLOS ONE's style requirements, including those for file naming. The PLOS ONE style templates can be found at https://journals.plos.org/plosone/s/file?id=wjVg/PLOSOne_formatting_sample_main_body.pdf and https://journals.plos.org/plosone/s/file?id=ba62/PLOSOne_formatting_sample_title_authors_affiliations.pdf

Response:Thank you very much for your suggestions. We apologize for the imperfections in the manuscript format that we submitted. We have made further corrections to the article's font, table format, page numbers, line numbers, and other details according to the requirements outlined in the PDF to ensure compliance with the journal's formatting guidelines. We kindly ask you to review and check it once again.

2.PLOS requires an ORCID iD for the corresponding author in Editorial Manager on papers submitted after December 6th, 2016. Please ensure that you have an ORCID iD and that it is validated in Editorial Manager. To do this, go to ‘Update my Information’ (in the upper left-hand corner of the main menu), and click on the Fetch/Validate link next to the ORCID field. This will take you to the ORCID site and allow you to create a new iD or authenticate a pre-existing iD in Editorial Manager.

Response:Thank you for your valuable suggestion. Following your advice, we have registered for an ORCID ID and have verified the corresponding author’s ORCID ID in the Editorial Manager. ORCID ID�https://orcid.org/0009-0008-8533-7097

MOE (Ministry of Education) Foundation on Humanities and Social Sciences [grant numbers 22YJA890032].

Response:Thank you for your suggestion. We have provided a more detailed explanation of the role played by the funders as per your request. “This study was supported by funding awarded to Professor Yang Jian from the MOE (Ministry of Education) Foundation on Humanities and Social Sciences [Grant Numbers 22YJA890032]. In his capacity as the funding recipient, Professor Yang Jian contributed to the study design, data collection and analysis, as well as preparation of the manuscript.” Due to our unfamiliarity with the operating system, we have included this statement in the cover letter and kindly request the editor’s assistance in updating this information. We sincerely appreciate your help once again.

4. We note that you have indicated that there are restrictions to data sharing for this study. For studies involving human research participant data or other sensitive data, we encourage authors to share de-identified or anonymized data. However, when data cannot be publicly shared for ethical reasons, we allow authors to make their data sets available upon request. For information on unacceptable data access restrictions, please see http://journals.plos.org/plosone/s/data-availability#loc-unacceptable-data-a

Response:Thank you for your guidance regarding data sharing policies. We fully endorse PLOS ONE's commitment to advancing open science and have carefully reviewed the journal's specific provisions on restricted data access (available at: https://journals.plos.org/plosone/s/data-availability#loc-acceptable-data-access-restrictions).

The data utilized in this study originate from the Population Health Data Archive (PHDA) (Identifier: CSTR:17970.11.A0031.202107.209.V1.0). In compliance with the database's mandatory security protocols:

Researchers must complete registration and pass eligibility review to obtain access rights; Approved users may analyze data exclusively within a secure virtual sandbox environment; Technical restrictions prohibit copying, downloading, or exporting raw data; Only processed aggregated results may be transmitted to researchers via email

These safeguards are implemented by the data custodians to ensure statutory compliance for sensitive information protection.

To proactively address your requirements, we have:

Explicitly stated in our Data Availability Statement:

"Study data are subject to PHDA access protocols and require registration/approval (Access link: https://www.ncmi.cn/phda/dataDetails.do?id=CSTR:17970.11.A0031.202107.209.V1.0)."

This access model aligns with established practices in peer-reviewed publications such as:

Li et al. (2024) in BMC Psychology (DOI: 10.1186/s40359-024-01719-4)

Zhao et al. (2024) in Journal of Public Health (DOI: 10.1007/s10389-024-02316-w)

Should further refinements to our approach be needed, we would be pleased to implement specific recommendations. We sincerely appreciate your professional dedication to balancing scholarly standards with data security.

5.Your abstract cannot contain citations. Please only include citations in the body text of the manuscript, and ensure that they remain in ascending numerical order on first mention.

Response:Thank you for your reminder. We have thoroughly checked and revised the abstract section of the paper to ensure that the current version of the abstract does not contain any references to citations.

6. Please update your submission to use the PLOS LaTeX template. The template and more information on our requirements for LaTeX submissions can be found at http://journals.plos.org/plosone/s/latex.

Response:Thank you very much for your constructive feedback. Please allow me to express my sincere appreciation for your time and effort in handling our manuscript and reviewing it.

We fully understand and respect the journal's strict requirements regarding manuscript formatting, as outlined in the PLOS LaTeX template. This is important for ensuring the consistency and professionalism of the publication. However, we would like to openly share some technical difficulties we have encountered. During the writing phase, we primarily used Microsoft Word, which is the tool we are most familiar with. After receiving your request, we immediately began attempting to convert our manuscript into LaTeX format, following the guidelines provided by your journal (http://journals.plos.org/plosone/s/latex) and utilizing the official template (plos_latex_template.tex).

Unfortunately, despite our efforts, we encountered unforeseen technical obstacles during the conversion process, mainly due to our limited experience with LaTeX syntax and compilation procedures. These challenges were particularly noticeable when dealing with complex tables, ensuring accurate citation formatting, and overall compiling and debugging. We deeply regret that we were not able to meet the formatting requirements on the first submission.

Considering the valuable time of both you and the reviewers, we respectfully ask for your understanding and kind assistance. We sincerely inquire whether it would be possible to proceed with the review based on our current Word document, as the scientific content is finalized and fully meets the requirements. We recognize the importance of formatting and assure you that once the manuscript receives a principal acceptance, we will dedicate all necessary resources to strictly conform to the PLOS LaTeX template within the specified time frame. We will seek professional assistance or engage in further study to ensure a successful conversion. Alternatively, would it be possible to offer any additional support? If submission in LaTeX is indeed a prerequisite for the review process, would the journal’s editorial team be able to provide some basic conversion assistance or a more detailed troubleshooting guide? We would be happy to provide the complete Word document and related materials to facilitate any possible support.

We fully acknowledge that this situation may add additional consideration for you and the reviewers, for which we deeply apologize. It is certainly not our intention to disregard the journal’s requirements, but rather we have encountered technical challenges. We greatly value the opportunity to publish in PLOS ONE and are committed to doing everything necessary to ensure that the manuscript ultimately fully complies with all formatting standards.

Whatever your decision may be, we fully respect it and will take immediate action on the next steps. If leniency cannot be extend

---

## [Decision Letter · Decision Letter 1]

19 Aug 2025

Recreational screen time and adolescent school adjustment based on latent profile analysis: the mediating role of perceived physical health

PONE-D-25-21675R1

Dear Dr. Li,

We’re pleased to inform you that your manuscript has been judged scientifically suitable for publication and will be formally accepted for publication once it meets all outstanding technical requirements.

Kind regards,

Javier Fagundo-Rivera, PhD

Academic Editor

PLOS ONE

**Additional Editor Comments:**

Dear Authors,

We are pleased to inform you that we have carefully addressed all the comments and suggestions provided by the reviewers. We believe that the revisions have fully satisfied the concerns raised, and the manuscript is now ready for publication.

Thank you very much for your time and consideration throughout this process. We greatly appreciate your work.

Kind regards,

Reviewers' comments:

Reviewer's Responses to Questions

**Comments to the Author**

Reviewer #1: (No Response)

Reviewer #2: All comments have been addressed

2. Is the manuscript technically sound, and do the data support the conclusions?

Reviewer #1: Yes

Reviewer #2: Yes

3. Has the statistical analysis been performed appropriately and rigorously?

Reviewer #1: Yes

Reviewer #2: Yes

4. Have the authors made all data underlying the findings in their manuscript fully available?

Reviewer #1: Yes

Reviewer #2: Yes

5. Is the manuscript presented in an intelligible fashion and written in standard English?

Reviewer #1: Yes

Reviewer #2: Yes

Reviewer #1: (No Response)

Reviewer #2: (No Response)

**Do you want your identity to be public for this peer review?** For information about this choice, including consent withdrawal, please see our Privacy Policy

Reviewer #1: No

Reviewer #2: No

---

## [Editor Report · Acceptance letter]

PONE-D-25-21675R1

PLOS ONE

Dear Dr. Li,

I'm pleased to inform you that your manuscript has been deemed suitable for publication in PLOS ONE. Congratulations! Your manuscript is now being handed over to our production team.

Kind regards,

on behalf of

Dr. Javier Fagundo-Rivera

Academic Editor

PLOS ONE